# Initiating interdisciplinary research for future-proof data protection in the context of Data Spaces and semantic interoperable data sharing.

Michiel Fierens[1]

[1] Centre for IT & IP Law KU Leuven, Sint-Michielsstraat 6 3000 Leuven, Belgium

### Abstract

The objective of this article is to propose a way forward for addressing the challenges currently facing data protection law in the context of a widespread implementation of Data Spaces and semantic interoperable data sharing. The failure to implement data protection law in an appropriate manner within this context could impede the implementation of Data Spaces and hinder the necessary protection of fundamental rights, as data protection can be considered a gateway right.[1] This is because the General Data Protection Regulation (GDPR) is based on assumptions and characteristics of data sharing that do not match the possibilities of Data Spaces and a broader evolution of semantic, interoperable data sharing. Although the call for future-proofing data protection has been heard for some time, it becomes even more relevant and tangible in a context of semantically interoperable data sharing. In this context, this article identifies the underlying assumptions and characteristics of data sharing on which current data protection law is based, contrasting these with the characteristics of semantic interoperable data sharing within Data Spaces. Subsequently, it identifies a series of key areas for further research, delineating common threads that can serve as a foundation for interdisciplinary discussions and research on future-proof data protection approaches in the context of Data Spaces and semantic interoperable data sharing. Moreover, based on these common threads, more specific preliminary suggestions for future-proofing data protection in the context of Data Spaces and semantic interoperable data sharing are also explored. In this way, the article contributes to the EU's objectives for the development of Data Spaces and benefits a wide range of stakeholders, including legislators, policymakers, enforcement authorities, providers and users of (personal) data spaces, and academics.

### Keywords

Data sharing, data protection, semantic interoperability, GDPR, Data Spaces

## 1. Introduction

The European Union has identified the necessity to improve the accessibility of data and knowledge within its European single market. This is based on the view that this will contribute to economic growth, competitiveness, and innovation. Ideally, innovation and competition no longer take place at the level of data collection or data availability, but at the level of service provision.[2] The aspiration to achieve this has constituted a significant driving force behind broader technical developments in the management and exploitation of interrelated data for a range of potential applications. For example, this can be observed with regard to the evolution of database systems in diverse data landscapes.[3] In consequence, the EU is now seeking to

---

✉ michiel.fierens@kuleuven.be (M. Fierens)

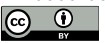 0000-0002-9721-2389 (M. Fierens)

establish environments or spaces where data from governments, businesses, and citizens can be securely shared and utilised in a way that fully exploits the potential interrelatedness between data.[4] Semantic interoperability is a pivotal factor in the establishment of these environments, designated as 'Common European Data Spaces'. By establishing a shared understanding of data definitions, structures, and relationships, data can be consistently interpreted across diverse organisational contexts.[5] The ability of computer systems to comprehend the meaning of data, and to establish or read connections between data, is facilitated by semantics. The overarching objective is to develop a system that is capable of semi-automatically identifying, establishing, enhancing and maintaining relationships between data and data sources within a specific context.[6] Accordingly, the concept of Common European Data Spaces encompasses a variety of elements and functionalities, designed to facilitate the discovery, integration and analysis of data originating from heterogenous sources. This, in turn, is intended to enhance data quality, accuracy and decision-making across a range of organisational contexts.[7] The objective is to move away from the prevailing approach to data sharing, which is based on the transmission of data in a specific syntactically interoperable format and requires a significant amount of data collection. This has resulted in the necessity for data to be sorted, labeled, and contextualized by each organization individually after it has been shared, in order to facilitate the extraction of meaning and value from the data.

The European Union (EU) is striving to enhance the accessibility of data and knowledge by exploiting the inherent interrelatedness of data in spaces where data from governments, businesses, and citizens can be securely shared and utilised. At the same time, it has enacted regulations to safeguard the fundamental right to data protection of individuals. These regulations have an indirect impact on the technology used to enhance data and knowledge accessibility, which may, in turn, constrain its potential. In this context, and given the current broad definition of personal data that triggers the applicability of the GDPR, this article focuses on the GDPR and personal data, rather than other legislation relating to purely non-personal data.[8] The advent of a more expansive evolution concerning data sharing provides an ideal opportunity to initiate interdisciplinary discourse on the existing challenges of data protection and to reflect on how these challenges can be further adapted in light of this novel and as-yet-unfolding evolution. This approach allows the EU's objective of achieving semantic interoperable data sharing and Data Spaces to be reconciled with the fundamental principles of data protection.

## 2. Semantic interoperable data sharing within Data Spaces remains underexplored by legal scholars

Although no universally accepted definition of a Data Space currently exists, it can be most closely described as an environment, defined by a governance framework and underlying technical infrastructure, which adheres to specific design principles set out by a given overarching organisation. The aim of such a "space" is to facilitate secure and reliable data transactions between organisations which are participating in that space.[9] The delineation of precise boundaries of the operation of a Data Space represents a challenging proposition, given the inherent ambiguity of the concept of a Data Space itself. To illustrate, a variety of Data Spaces may exist as standalone Data Spaces or as components of a more comprehensive Data Spaces. In this regard, any definition is necessarily broad, as it must encompass the potential

for data sharing in a variety of environments. Moreover, the further development of functionalities and semantic interoperable data sharing in these environments will occur in an incremental manner, evolving in accordance with time and the specific requirements of the context.[10] Over time, more comprehensive functionalities and methods for the facilitation of the discovery, integration and analysis of data across a diverse range of storage locations and data types will be developed in such environments. These functionalities will enable the seamless further sharing and utilisation of data through the exploitation of the interrelatedness between data and the extraction of knowledge from it upon sharing.

Nevertheless, the concept of semantic interoperability, which is pivotal to a new approach to data sharing, as well as the development of these innovative data sharing environments in which this form of interoperability is used, has not been sufficiently addressed by legal scholars. Although interoperability has been studied in consumer and competition law, no distinction has been made between a syntactic (where computer systems do not understand relationships between data) and semantic context of data sharing. The objective of these studies was mainly to examine the potential for interoperability to achieve the objectives of both branches of law. [11][12][13] Therefore, it failed to consider the potential to facilitate the discoverability and usability of data across organizational boundaries, as well as to enhance the exploitation of relationships between data through the use of for example data models. In the context of data protection law, research has predominantly adopted a relatively limited approach to data sharing. This approach has typically involved allowing organisations to determine the appropriate structure for sharing data for further copying, sorting, labelling and contextualising within their own environment. The concept of interoperability was primarily regarded as a mere instrument for operationalising the right to data portability and facilitating data transfers between different data silos. In this context, the authors considered closed, static environments in which personal data is shared using only a standardised data format. Nevertheless, as will be discussed in the following sections, the potential for providing functionalities in a diverse landscape in terms of data sources and structures used through open, dynamic environments and the use of semantic interoperability was not yet a consideration.[14][15][16] Furthermore, the current shortcomings in data protection were not taken into account in the context of these prospective developments. Lastly, there is even research on federated learning[17][18][19] and decentralised data processing[20][21]. In the context of providing functionalities and consequently processing data across distributed repositories of data, this can be a useful approach. Instead of being trained on a central server, machine-learning models are trained on local or decentralised storage places, like a party's device. This can be partially compared to the use of Data Spaces and semantic interoperable data sharing, whereby functionalities are combined to exploit the interrelatedness between data and thus extract knowledge from it in an environment with various distributed or decentralised storage locations. It is important to note, however, that semantic interoperability, namely the exploitation of relationships between data by enabling computer systems to better understand the meaning of data through for example data models, can also serve as a key enabler of machine learning. The advancement of semantic data models, which provide structure in diverse data landscapes, has the potential to enhance the predictive power of machine learning methods.[22] Moreover, the ongoing development of this technology and its implications for data protection have yet to be fully identified or acknowledged.[23][24]

# 3. Research objectives and methodology used

## 3.1. Research objective

The objective of this article is to provide an initial introduction to the broader evolution in data sharing for legal scholars who are not already familiar with it, as well as to offer insights into the potential developments that may further occur in this field. In particular, the article aims to examine the potential of utilising semantic interoperable data sharing in combination with the proposed Data Spaces by the European Union, and its implications for data protection legislation. It is recognised that, in pursuit of this aim, certain elements pertaining to semantic interoperable data sharing and Data Spaces may be simplified to a degree that is not in exact alignment with the technical specifications. However, a balance is always sought between the use of technical and legal terms. Once the characteristics of the future evolution in data sharing are exposed, the current shortcomings in data protection can identified as well. In this way, the identified shortcomings can immediately be considered in the context of the broader future evolution of data sharing. This should prompt an interdisciplinary debate and research initiatives aimed at developing future-proof data protection strategies. The following sections in the article demonstrate how the use of semantic data models and the implementation of several functionalities in Data Spaces shift the emphasis from data collection to the provision of new data-intensive services and render identified and existing data protection issues even more pertinent and tangible, while also introducing novel specific challenges.

## 3.2. Broader evolution in data sharing and consequences for data protection

The EU has made the creation of Data Spaces and the use of semantic data interoperability as its fundamental building block a major policy choice to facilitate (personal) data sharing in the European Single Market. The fundamental importance of semantic data interoperability for facilitating a new way of sharing data already emerged clearly in the public sector through the establishment of the European interoperability framework[25], and the proposal for an Interoperable Europe Act.[26] The objective of Data Spaces, as well as that of semantic interoperability, is to improve the services offered by organizations. Rather than having to deal with large and diverse amounts of potentially useful and interrelated data, which may be unevenly structured and thus difficult to search and make use of, such an environment should be equipped with the capacity to facilitate a spectrum of data analysis services and provide core functionalities that are indispensable for the effective retrieval and use of valuable data. To achieve semantic interoperable data sharing in Data Spaces, data are mapped into semantic data models. Simply put, data and their interrelationships, as well as further information about that data are thereby mapped into such a semantic data model so that a computer system can read and understand it.

However, there is considerable diversity of opinion among organisations as to the optimal approach to establishing Data Spaces and implementing semantic interoperability in such an environment. In particular, the methods employed to exploit relationships between data and, consequently, represent knowledge (data models, vocabularies, etc.) upon sharing vary depending on the context and the aim of the Data Spaces. It is crucial to highlight the incremental nature of the broader evolution in data sharing, where a balance must be struck

between uniformity and the accommodation of disparate organisational requirements and varying data infrastructures. Consequently, the extent to which relationships between data can be exploited and knowledge can be represented and extracted from datasets, thus facilitating semantic interoperable data sharing, may be more limited in practice.[27][28] The non-standardised use of data models and vocabularies also poses challenges.[29] More specifically on the technical side of semantic interoperability, there are still challenges in terms of knowledge extraction or the derivation of insights[30] as well as flexible and advanced querying[31] of semantic interoperable and thus interconnected data.

The above considerations must be borne in mind throughout the article. Given the inherent difficulty in achieving a balance between uniformity and the accommodation of disparate organisational requirements and varying data infrastructures, this article does not seek to provide an exhaustive overview of potential ways to set up Data Spaces. Furthermore, it does not analyse all the implications of data sharing through Data Spaces on different legal frameworks. This article, rather than focusing on a specific aspect of data sharing, aims to provide an overview of the broader evolution of data sharing that is being driven by the use of Data Spaces and semantic interoperability. These two concepts, in combination, establish various basic functionalities that facilitate new ways of processing data across a diverse landscape of different storage locations and data structures. The widespread use of semantic data models and the dynamic and collective characteristics of Data Spaces deviate from the assumptions underlying the current data protection framework, as will be demonstrated in the following sections. In that regard, the article begins by providing a brief overview of the assumptions that underpin the GDPR, before translating these into concrete problems concerning the application of the GDPR in new technological contexts. This illustrates how the broader evolution of data sharing makes current problems even more relevant and tangible. In this way, the article aims to make a concrete call for interdisciplinary debate, while also identifying potential avenues for further research.

## 3.3. Research design and methodology

The objective of this article is to establish a foundation or a way forward for future research into future-proof data protection in the context of Data Spaces and semantic interoperable data sharing. It could be argued that data protection represents a gateway right to the respect of other fundamental rights of data subjects.[1] It is therefore of the utmost importance that, in light of the extensive realisation of Data Spaces and semantic interoperability, current data protection issues are not reinforced or even intensified, as this could have a significant impact on the fundamental rights of data subjects.

Firstly, the primary characteristics of semantic interoperable data sharing within Data Spaces will be identified and explained in order to facilitate a comprehensive understanding of the broader evolution regarding data sharing. This article then builds on the aforementioned characteristics and subsequently demonstrates the shortcomings of the underlying assumptions and characteristics of data sharing as set forth in the GDPR in that context. In light of these characteristics and shortcomings, the article proceeds to propose several key research areas that should form the basis of any future research on new data protection approaches. These key research areas can be seen as common threads, which serve to provide a framework for further interdisciplinary research into Data Spaces and semantic interoperable data sharing. Moreover, these common threads are situated within the context of existing research on future-proof data

protection approaches. Consequently, future research can build upon them while also distinguishing itself from existing research in light of the specific characteristics of Data Spaces and semantic interoperability, as outlined in Section 4 below. In conclusion, the paper also considers potential preliminary suggestions regarding the adaptation of the prevailing underlying assumptions of the GDPR, based on the aforementioned common threads.

It is recommended that new technological developments incorporate data protection from the outset in their design. Nevertheless, pursuing such a design has become an exceptionally challenging endeavour. The abstract nature of such a fundamental right, as well as the normative regulatory framework that protects it, presents a significant obstacle in applying these interpretations in specific contexts. Given the pioneering nature of semantic interoperability and Data Spaces, the Designing-by-Debate (DbD) method is deployed as the principal methodology for delineating the characteristics of this new evolution, as well as its implications on the underlying assumptions and characteristics of data sharing as enshrined in the GDPR.[32] The value of this approach lies in its capacity to integrate the perspectives of stakeholders from a range of areas of expertise. Software engineers may, for instance, adopt a narrow perspective on certain concepts, such as data protection. In contrast, lawyers possess a more comprehensive understanding of these concepts, coupled with an awareness of the potential implications of new technological developments on other fundamental rights. In accordance with a DbD approach, a broader societal perspective is intrinsic to the design of any given research project. Consequently, this approach enables the research questions to be solved by integrating the perspectives of stakeholders from different areas of expertise. In this context, and for the purposes of writing this article, SolidLab Flanders, which provides financial support to the author of this article, represents an exemplary case in which a DbD approach is being employed to investigate broader societal challenges. The consortium is comprised of stakeholders from a range of disciplines, including computer science, law, economics, and communication sciences. Collectively, they are engaged in exploring the potential of personal data spaces and their application in the data economy in Flanders. This exploration is conducted in collaboration with policymakers, citizens, and entrepreneurs in a quadruple helix framework. Participatory exercises facilitate the mapping of views and practices among relevant stakeholders from different areas of expertise. In this way, consequences that were not foreseen for individuals, industry and society alike are brought to light, and the normative issues that they raise are identified.

## 4. Characteristics of semantic interoperable data sharing

### 4.1. Functionalities distributed over heterogenous, scattered storage locations

Similar to the early days of the Internet and the absence of search engines to look up websites more specifically, the European single market used to be nothing more than a collection of separately accessible databases or data storage locations. As computer systems become capable of understanding the relationships between data, through the use of semantic interoperable data sharing and the use of semantic models, opportunities have arisen to facilitate a spectrum of data analysis services and provide core functionalities such as browsing, querying and cataloguing data that are indispensable for the effective retrieval and use of valuable data in environments where data is unevenly structured and thus difficult to search. The growing use of semantic data models and the creation of interconnected infrastructure (named middleware

or connectors) through the use of equipment and design principles provided by organisations establishing principles and specific (modular) software for Data Spaces such as IDSA, Gaia-X and FIWARE enables organisations to semi-automatically identify, establish, enhance and maintain relationships between data and data sources within a specific context. In this respect, it can be considered to be comparable to the establishment of an integrated database that is accessible, searchable and comprised of the data from a number of different existing databases.[33] In the past, achieving this result necessitated the consultation of each database individually, followed by the additional processes of structuring, labelling and contextualising the data retrieved from that database. Parties immediately obtain knowledge and value by searching through a mixture of different types of storage locations.[34] The location of the data storage is of lesser importance, as a large, searchable network comprises a number of disparate storage locations, interconnected by Data Spaces and the deployment of semantic data models to facilitate computer systems' ability to exploit the interrelationships between data.[35] While this broader evolution in data sharing also offers opportunities to implement sovereignty mechanisms over data storage locations through Data Spaces and semantic interoperability, the potential risks lie in the fact that connecting elements and functionalities are provided on top of this diverse and complex data landscape. This facilitates novel approaches to data processing, as well as more sophisticated forms of processing, given that semantic data models permit computer systems to interpret relationships between data upon sharing. The precise impact of those connecting elements and functionalities such as for example brokering services, cataloguing data sets and common services, as well as providing specific semantic data models or vocabularies for mapping data, remains underexplored.

## 4.2. Leveraged interrelatedness of data

From a social perspective, certain data is inherently relational, frequently pertaining to friends and family. It can thus be argued that the processing of data from one individual entails the processing of data from other individuals, who may be considered data subjects in their own right.[36] In this context, it is interesting to note that Data Spaces and the use of semantic interoperability facilitate the expression and subsequent processing of relationships between data. In light of the fact that different types of data with varying structures from a multitude of data providers within a Data Space can be retrieved in a manner that facilitates ease of association with other data available in the Data Space, the distinction between personal and non-personal data, as well as that between the so-called special categories of data, such as sensitive data, becomes increasingly difficult to maintain.[37] It is relatively straightforward to classify data sets that contain both personal and non-personal data as being 'inextricably'[38] linked to each other[39]. The advent of Big Data and novel data analysis techniques within closed silo environments has already rendered the existence of these categories questionable. However, the more widespread use of Data Spaces and semantic interoperability, where a spectrum of data analysis services and functionalities in environments with dynamic and collective characteristics enable the further use and further exploitation of potentially interrelated data for every potential Data Space participant, makes these categories no longer tenable at all.

## 4.3. Dynamic and collective data sharing environments

In a big data or data silo context, the collection and subsequent extraction of knowledge always involved copying, sorting, labelling and contextualising the (received) data. This has always been done within closed environments managed by a limited number of parties.[40] However, the European Union aspires to eliminate multiple copies of data and new closed environments. To achieve this, the EU advocates the 'once only' principle in data spaces with semantic interoperable data sharing. When data is made available in that Data Space to a particular organisation, it should be retrieved from its original storage location(s) and re-used without making copies when possible.[41] In this regard, the advancement of various sovereignty and trust mechanisms, such as the capacity to encapsulate semantic interoperable data in a Data Space with policies for its subsequent use, is intended to address concerns pertaining for example to intellectual property, which are associated with this reuse. However, these concerns will not be further elaborated upon in this discussion.[42]

Data Spaces making use of semantic interoperable data sharing are inherently dynamic. After all, semantic data models, vocabularies, data catalogues and data analysis services are constantly being refined by multiple parties involved in a Data Space.[43] For example, according to a specific context or need, certain data sets can be made available in the central catalogue of the Data Space, after which the parties themselves can select a semantic data model in which to map that data set, rework it and make it available again in the Data Space. Indeed, these parties can also further refine the semantic data model and the exploitation of the interrelatedness of the data in their specific context. This approach permits the establishment of even more comprehensive relationships between data, thereby facilitating the enhancement of knowledge extraction. Data Spaces are consequently seen a dynamic and collective effort.

The collective and dynamic aspects can be clearly highlighted by contrasting the visual representation of the data life cycle in a semantic interoperable data sharing ecosystem with that of the classical data life cycle. The classical cycle is depicted as a straight line, representing a closed environment with limited parties involved. It starts with the creation of data and ends with its destruction, and then a new cycle begins. In contrast, the data life cycle used in Data Spaces with semantic interoperable data sharing is continuous and has no end. In this cycle, each party can contribute at any point, and the inputs from different parties build on one another (Figure 1).[44] The collective in a semantic interoperable environment such as a Data Space, as it were, manages the data and its life cycle.

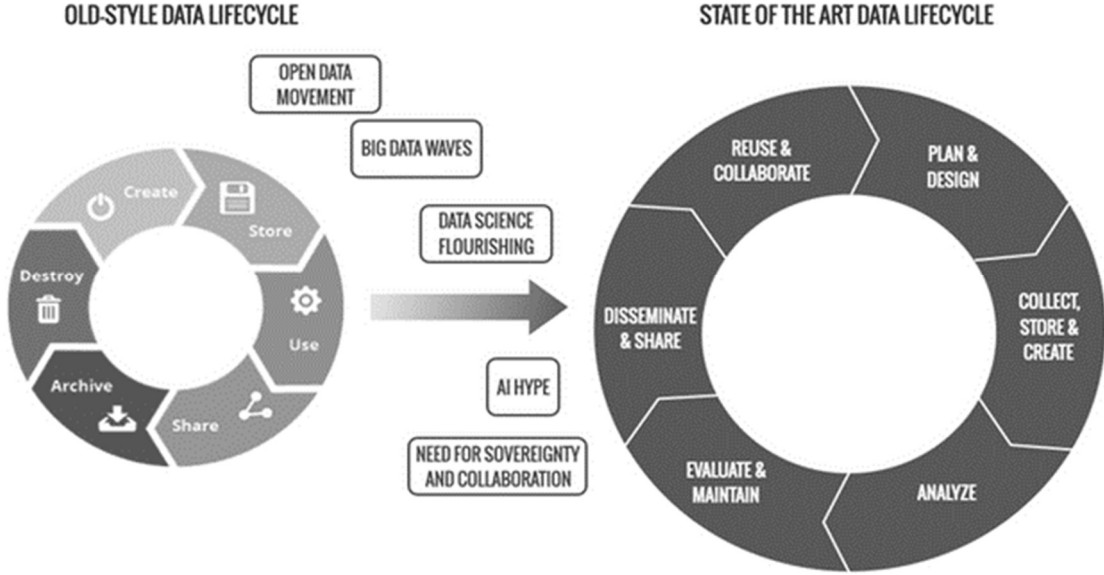

**Figure 1:** Data Lifecycle 'evolution' as advance by BDVA.

# 5. Underlying assumptions and characteristics of data sharing under the GDPR

The General Data Protection Regulation (GDPR)[45] is based on the fundamental right to the protection of personal data as stated in Article 8 of the EU Charter of Fundamental Rights. The GDPR permits the sharing and processing of data, provided that certain principles are observed. In addition, individuals (data subjects) are afforded the right to exercise forms of control over their data, for instance, by exercising their data subject rights.[46] The GDPR remains a crucial tool to maintain a balance between sharing personal data and protecting individuals' data in the context of new technological developments in information collection and data sharing. Nevertheless, in light of the characteristics of Data Spaces and semantic interoperable data sharing outlined in the preceding section, it becomes evident that the underlying assumptions and characteristics of data sharing under the GDPR are not without shortcomings.

## 5.1. Taking into account closed and static environments for data sharing

The GDPR is regarded as a forward-thinking and technology-neutral framework that addresses emerging technological advancements in data collection and sharing (recital 15 GDPR). Nevertheless, at the time of the drafting of the GDPR, the EU was particularly aware of the data silo structures employed by Big Tech companies, which were attempting to limit data sharing with other companies in various ways.[47] The initial step toward enabling data sharing across closed systems owned by different parties was the development of a common, machine-readable format so that data can be more easily transferred from one company to another.[48] When considering the right to data portability in the GDPR, initial proposals indicated a potential for the use of standards regarding semantic data models. However, the EU ultimately opted for a less ambitious formulation and requirement of standards regarding interoperability, which

resulted in a focus on formal aspects of data sharing and thus on organising data according to pre-determined instructions[49]. The right to data portability shows that the EU focused mainly on enabling data subjects to receive their personal data in a machine-readable, structured and commonly used format. Data subjects could then take their data in this format to another company, where it had to be labelled, contextualised to be able to extract knowledge from it and further use it in a meaningful way.

## 5.2. Focus on data as input of the data processing operations

The GDPR currently starts from the idea that the nature of data is the focal point for the application of data protection legislation. So not every processing of any kind of data can be covered by the GDPR, only data relating to identified or identifiable people.[50] In addition, processing certain types of data, sensitive data such as health data, also requires taking additional safeguards. The European legislator's decision to focus on controlling objectively defined categories of data, such as personal and sensitive data, can also be explained by the origins of data protection law in information theory. Information theory is based on the use of mathematical principles to ensure the effective transfer of data between communication systems and is consistent with a specific focus on the sharing of data between closed and static environments.[51] In this context, the data itself, or in other words the input to the data sharing process, is essential. Data analysis services and functionalities across a diverse landscape of data storage locations are not considered. However, given the potential of Data Spaces and semantic interoperability to exploit the interrelatedness of data, as outlined in Section 4.2, it is challenging to develop a comprehensive theory that distinguishes between the legal status of different categories of data.[52]

## 5.3. Underlying principle of separation of data processing operations

The purpose limitation principle represents the central nexus of data processing throughout the GDPR. It emphasises the importance of specifying the purpose of data processing as an initial balancing exercise before any data processing takes place. In order to mitigate the risks associated with the processing of personal data, it is essential that personal data is only processed within the context and framework of the specified purpose. Moreover, it ensures that data processing is transparent and predictable for the data subjects concerned. Data controllers (the party determining the purposes and means of processing following the GDPR) may not link data collected for different purposes to one broad purpose and process them only, for example, because their use might be advantageous in the future. A comprehensive interpretation of the GDPR, which implies reading it in a holistic way, for example by considering all other principles and obligations in the context of purpose limitation, shows that it should in principle be prohibited to integrate different data (sets) that were originally processed for other purposes. In particular, the principles of storage limitation, data minimisation (Article 5 GDPR) and the obligation of data protection by design and default (Article 25 GDPR) require that collected data are stored or at least only accessible separately according to their different purposes. In this context, Felix Bieker identifies an underlying objective of separating processing operations, thereby also separating the storage and subsequent utilisation of data.[46] This can be reframed in the original context of closed, static environments, where data was separately labelled and contextualised for further use, and the

emphasis was on the concept of data itself as input. The dynamic and collective manner in which data is shared and subsequently used, integrated or refined in Data Spaces through an array of data analysis services and functionalities and thus processing operations, initially appears to be at odds with the principle of separation.

## 5.4. Emphasis on the individual's ability to manage personal data

Another aspect that can be read into the GDPR is the emphasis put on respecting autonomy and human dignity of data subjects while processing personal data.[53] It can be argued that data protection is primarily concerned with a self-determination approach, whereby the individual's ability to exercise a form of control over their personal data processing is of paramount importance.[54] The initial focus is on the individual and their capacity to comprehend the specific circumstances surrounding the processing of their personal data.[55] For example, the legitimacy of processing (specific) categories of personal data under the lawful processing ground of consent or explicit consent hinges on the individual's own responsibility to accept this as a lawful processing ground for the data controller. Further, the concept of data protection rights is founded upon the fundamental values of autonomy and human dignity; thus, they are primarily associated with the data subject's capacity to play a role in the process of data protection. The right to data portability and the right to access, which serve as a nexus for all other data subject rights, build on the idea that individuals (data subjects) can exercise a form of control over their data and are able to manage their own data.[56] In doing so, data subjects should be able to receive their personal data in a machine-readable, structured and commonly used format in order to share data freely between service providers (and consequently data controllers).[57]

Nevertheless, the individual's ability or responsibility can be questioned, given that it is not uncommon for an information asymmetry and power imbalance to persist between a controller and an individual as a data subject in such a context.[58] Moreover, historically, data protection has never focused solely on protecting individuals. Over time, this has become the dominant narrative.[59] A predominant focus on the data subject's ability to make his or her own data protection decisions, and thus on the individual, does not take into account the broader history of data protection law and the role that the collective plays in it.[60] In light of the collective and dynamic environments that Data Spaces and semantic interoperability entail, it is challenging to justify the proposition that the principles set forth in the GDPR should be primarily examined from the perspective of the individual.

# 6. Towards future-proof data protection approaches

This section identifies key research areas that can provide a foundation for further interdisciplinary research into Data Spaces and semantic interoperable data sharing. These key research areas are based on existing research on data protection approaches in new technological contexts. In this respect, they can serve as a foundation upon which future research can build, while also distinguishing itself from existing research. The relevance and tangible nature of these key research areas is further enhanced by the advent of semantically interoperable data sharing and Data Spaces. Indeed, these key research areas can be regarded as common threads which should be taken into consideration with regard to research aimed at establishing a balance between the underlying assumptions and characteristics of data sharing

in accordance with the GDPR and those of Data Spaces and semantic interoperable data sharing. In this context, the article also explores preliminary suggestions regarding potential adaptations to the GDPR in light of the specific characteristics of semantic interoperability as set out in Section 4.

## 6.1. Division of responsibilities

The advent of Data Spaces and semantic interoperable data sharing is giving rise to the emergence of new data sharing ecosystems, with new types of services and responsible parties and even a potential whole collective responsibility for a Data Space itself. In addition, the growing modularity and, consequently, interoperability of software components facilitate this process and diversity. With each incremental development, an additional layer of complexity has been introduced over time. The fact that the knowledge extraction and a broader semantic evolution in data sharing occurs in dynamic environments, with collective efforts being built upon, serves to increase the complexity of the situation. This is particularly relevant in light of the current extensive and vague jurisprudence concerning personal data and the concept of data controllership. The broad interpretation of joint-controllership under the GDPR in the current case law, as well as the many grey areas surrounding this concept of controllership, impede the predictability of the precise responsibilities and, in turn, complicate the effective implementation of the GDPR. Almost anyone can qualify as a controller in that respect, so to speak.[61]

Similar problems have already been extensively described in the literature, for example in relation to cloud computing and accountability in distributed or decentralized environments (e.g. with scattered storage locations). There, too, complex technical infrastructure chains ensure that multiple parties play a role in the data processing.[62] A conclusive solution regarding the application of the GDPR in such complex chains with different parties, however, has not yet been found. It follows that the expansion of joint control gives rise to the necessity for active collaboration. Nevertheless, no court, nor the Article 29 Working Party or EDPB, offers guidance on potential default scenarios of coordination or further specifications of what should occur in the absence of such coordination.[63] This leaves room for the designation of so-called accidental controllers. Those with actual influence over the purposes and means of processing may derive benefit from such ambiguities and may seek to transfer their obligations to other actors.[64] Such a situation may even give rise to a paradoxical effect, resulting in a lack of accountability. In that regard, the subsequent proposal of a step-based approach to reduce the complexity of the division of responsibilities had the unintended consequence of introducing an additional layer of complexity. [63] In consideration of the step-based approach, there appears to be a shift in focus towards a microscopic view of the processing operations. Nevertheless, there is a notable absence of guidance regarding the extent to which the division of responsibilities of the parties in question could be balanced in such a situation. Furthermore, there is no examination of the broader implications and thus an additional macroscopic view. In addition, it is noteworthy that no data protection authority makes reference to the possibility of differentiating between the various forms of enforcement that could be applied to parties jointly responsible.[65]

In this regard, existing literature proposes, for instance, the narrowing of the controllership scope and the imposition of a higher threshold for the level of influence over processing means. [64] It is noteworthy that the technical implementation of semantic interoperability can

facilitate the determination of and subsequent narrow delineation of controllership and even its subsequent translation in practice.[66] To illustrate, in a semantic interoperable data sharing environment, such as a Data Space, data or knowledge flows can be formalised in semantic data models in a manner that facilitates the development of logging mechanisms to ensure transparency and data provenance during the processing of data. This allows for the determination of controllership.[67] In order to achieve this objective, efforts have already been made to translate the obligations set out in the GDPR into machine-readable computer language. This allows the legislation to be linked to the use of data in such environments, thus facilitating its implementation.[68] In that regard, the Open Digital Rights Language (ODRL) allows data and metadata to be modelled in such a way that compliance of certain parties with the GDPR can then be automatically verified.[69]

## 6.2. Collective interests

The collective and dynamic context that Data Spaces and semantic interoperable data sharing create suggest a greater focus on collective interests. In this context, the organisation of Data Spaces will be contingent upon the specific context and the particular organisational requirements of the specific use cases in question. This will entail the achievement of a collective purpose through the use of semantic interoperable data sharing within the Data Space. Referring to fundamental European values, one should in that regard look beyond individual autonomy to the outward-looking dimension and the relationship of one's choices to those of other individuals and collective interests.[70]

In a similar vein, existing principles and rights of the GDPR could be interpreted in a more collective way for the purpose of data sharing through Data Spaces.[71] The GDPR's right to portability, for example, allows the interests of others to be taken into account when exercising that right. Similarly, the principles of data minimisation and accuracy can both be interpreted simultaneously at the individual and collective levels. For example, at the individual level something may be accurate, but at the collective level, with respect to multiple data subjects, more data may still be needed to provide an accurate representation with respect to the collective. Even the interpretation of a purpose can be expanded more collectively to encompass the entire Data Space, as long as adequate safeguards are in place. This would mean that specific processing operations can only occur within the overarching purpose defined at the Data Space level. However, without additional changes regarding the division of responsibilities, this approach may also provide a means for controllers to avoid their responsibility, as they can always refer to the fact that the Data Space (operator) itself determines the purposes of processing and thus retains control over the processing. Furthermore, at this time, the potential trade-offs between the GDPR's principles and rights[72], as well as between an individual or collective interpretation[73], are not explicitly or clearly delineated in guidelines. Further research is also required in order to gain a greater understanding of the matter in question.

Lastly, the collective, as it were, manages the data and its life cycle and consequently the interconnectedness of (personal) data. The collective and long-term impact of semantic interoperable data sharing should therefore be considered. The new Data Governance Act[74], relies on data intermediaries[75] such as data cooperatives and data altruism organisation to collectively manage data and the interests of multiple data subjects.[76] One possible avenue to pursue is to permit individuals or data intermediaries to define access conditions for the use of personal data through the technical possibilities inherent in semantic interoperability.[77]

However, overly paternalistic treatment of the collective may also overly harm individual autonomy. Future research should therefore thoroughly examine in which cases (in which contexts) individual interests still matter and outweigh possible collective interests.[53]

## 6.3. Broader data-related harms

Authors have also pointed towards protection against so-called information-related harms that concern both the collective and the individual.[78] These kind of harms arise specifically in situations where a party is able to exploit the interrelatedness of data sets through the use of semantic models and subsequently extracts knowledge from acquired data or information which does not necessarily qualify as personal data (e.g. because it aggregates large quantities of data) and, as a result, falls outside the scope of the GDPR. In that regard, the complete implications of Data Spaces and semantic interoperable data sharing, particularly in regard to information-induced harms, remain uncertain.

Current tools provided in the GDPR such as Data Protection Impact Assessment or Data Protection by Design have the potential to tackle these harms. In practice, however, they do not adequately address these.[79] Lawmakers hoped to address as many unwanted side effects of digitisation as possible with the GDPR.[80] In practice, on the other hand, policymakers are not always eager to legislate abstract and sometimes difficult to foresee collective or information-related harms, for example due to a lack of precise falsifiability.[81] Consequently, the GDPR affords data controllers greater freedom to make assessments regarding potential information-related harms and places responsibility on data subjects to object. A shift in focus from the regulation of information or data (input of the process) to the regulation of knowledge (output of the process), for instance through the consideration of broader social impact assessments or ex ante risk-based government prohibitions of data use, is a compelling argument when discussing future-proof data protection approaches in Data Spaces.[82] Once more, data intermediaries may be instrumental in protecting such broader societal interests and in facilitating broader societal impact assessments for further data use. A review of recent legislative initiatives, such as the AI Act[83] and the Digital Services Act[84], also indicates the potential for risk-based government prohibitions to mitigate broader risks and harms in society in advance. Similarly, future research could investigate the potential for extending the application of certain data protection principles to encompass computer code more broadly. The advancement of semantic models and vocabularies in accordance with these principles may serve as a means of mitigating potential information-induced harms.[85]

## 7. Conclusion

The GDPR's existing assumptions and characteristics of data sharing ignore how data sharing is evolving in relation to the wider use of Data Spaces and semantic interoperability. The use of semantic interoperable data sharing has enabled the provision of diverse data analysis services across heterogeneous environments where data was previously characterised by inconsistency in structure. Furthermore, it allows organisations to promptly exploit the interrelatedness between data and data sources within a specific context upon sharing, thus facilitating immediate knowledge and value extraction. With the EU now fully committed to creating European Data Spaces and the use of semantic interoperability, the consequent creation of consortia and bringing together stakeholders provides the perfect opportunity to develop

future-proof data protection approaches. Therefore, this article provides a foundation for future interdisciplinary research into such approaches. It outlines common threads that should be followed in such research. These threads build on existing research on concerns related to the application of the GDPR in new technological contexts. However, the common threads become even more relevant and tangible with the creation of Data Spaces and the associated use of semantic interoperable data sharing.

In addition, this article also explores preliminary suggestions for future-proof data protection approaches in light of the specific characteristics of semantic interoperability. Firstly, the advent of complex data-driven supply chains necessitates the development of a more nuanced understanding of the interrelationships between the different actors involved. In the event of multiple parties being held responsible under the GDPR, it is essential that a set of guiding principles is in place to ensure a fair and comprehensive division of responsibilities. This must take into account a macroscopic view of the processing chain. Once the responsibilities in question have been legally delineated or reinterpreted, technical semantic interoperability capabilities can facilitate the translation of such a delineation or reinterpretation in practice via logging mechanisms and automatic verification of compliance with several data protection obligations. Second, the European Data Protection Supervisor could propose more collective interpretations of the principles and rights in the GDPR, and data protection authorities could also enforce them in a more collective way. In this respect, a new balance between individual and collective interests needs to be found. Data intermediaries in the new Data Governance Act provide an interesting ground for further research in this regard. Once more, the technical possibilities inherent in semantic interoperability could enable data intermediaries to embed access conditions in the data they manage for data subjects, thus facilitating the automatic and balanced conditioning of further use of that data. Thirdly, the consideration of broader societal harms and the shift in focus from the regulation of data to the regulation of knowledge processing and the associated increase in information-related harms are important aspects for further discussion. In this context, future research could investigate the necessity of certain limitations on specific forms of knowledge extraction or data reuse. Furthermore, the potential benefits of broader societal impact assessments when processing data should be further examined. The utilisation of semantic data models and vocabularies could also be subjected to certain fundamental data protection principles and consequently design requirements.

## Acknowledgements

Michiel Fierens is supported by SolidLab Vlaanderen (Flemish Government, EWI and RRF project VV023/10).

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
