# OpenReview forum: "Initiating interdisciplinary research for future-proof data protection in the context of Data Spaces and semantic interoperable data sharing."
_SEMANTiCS.cc/2024/Workshop/NXDG — NXDG 2024 Conditionallyed_

### Official Review · ~Lola_Montero_Santos1 · 2024-07-23
**Good Contribution**

**Rating:** 8
**Confidence:** 3

**Review:**

The article is very interesting and provides a useful and constructive assessment of the GDPR vis-a-vis semantic interoperability.

**Comments on improvements**

* The reading of the text is not always easy. There are some grammar and readability issues. For example:

> “This can be compared to use of semantic interoperability in for example Data Spaces, whereby data
is extracted from various distributed or decentralised storage locations and aggregated into
knowledge.” (pg. 3)

>  “Data Spaces should not function as separate silos with isolated data and knowledge, but instead
should be searchable by authorised and consist of linked data originating from multiple (often
distributed or decentralized) storage locations.” (pg. 3)

> “More specifically on the technical side of semantic interoperability, there are also still challenges in
terms of knowledge extraction or the derivation of insights[29] as well as flexible and advanced
querying[30] of semantic interoperable and thus interconnected data.” (pg. 4)

* The title is too bold. The article is much more constrained than “The Way” to future-proof data protection.

* The scope of the analysis in this article (“the potential impact of the implemented policy
on the existing assumptions and characteristics of data sharing as outlined in the GDPR”) contrasts with the many comments made on European Data Spaces. The author should explain how this analysis is relevant for European Data Spaces.

* What are “suboptimal legal choices”? “According to whom are these legal choices suboptimal? (pg. 4)

* What is the “Designing-by-Debate method (DbD)” (pg.4). More info should be given on this in the main text. SolidLab Flanders is mentioned without providing any information of the initiative. What does SolidLab Flanders do? How do they fit in the main thread of the article? The reader may not know.

* “The use of data is no longer constrained to the precise storage location, or in
other words, no longer implies storage at the party itself and its subsequent application by the data's user.” (pg. 4) Do you mean user’s data?

* Section “4.3. Open and dynamic data sharing environment” radically contrasts the behaviour of most private firms and IPRs regimes. The author should somewhat address this. Or does the author only refer to Public Administrations? Also, this section appears to base itself on utopic objectives. The open and dynamic data sharing environments are a policy objective, they have not been realised. This should be acknowledged, or real examples provided.

* What about when a party in a Data Space decides to change the metadata or break out of the data space? How is this “State of the art data lifecycle” resilient? (pg. 6-7)

* I really do not understand the following paragraph on pg. 10: “While it is important to keep in mind that the combination of the genuine re-use of data, the increasingly aggregated nature of data processing and the improvement of knowledge extraction, the need for the collective in dynamic data management, and the interconnectedness of data are characteristics of semantic interoperable data sharing that have not yet been fully taken into account, such prior work can nevertheless provide a starting point for further reflection on future-proof data protection approaches.”

* VERY IMPORTANT. In the Data Act, the user holds the ultimate decision on whether to share or not a data point. I do not think that the author’s suggestion that “the Data Act's requirement that private sector entities share certain types of data more widely in order to promote collective economic growth” is true (page 11). I have checked the Footnote (73), and the author is quoting an outdated document: the proposed Data Act, not the adopted Data Act.

---

### Official Review · ~Rigo_Wenning1 · 2024-07-30
**Pourquoi faire simple si on peut faire compliqué.**

**Rating:** 4
**Confidence:** 4

**Review:**

First of all, the PDF submitted is a graphics, not a text PDF. This is an additional burden for review. Given that latex was used, one can openly question the intent by submitting graphics instead of text. Therefore, this review will not cite text from the article and the author will have thus to find the mistakes himself.

Up to page 5 the author just explains that he makes assumptions and that the articles doesn't do this and that. The definition of "Data Space" is omitted, so everything turns into a Data Space. Some assertions are not supported by the underlying citations. I verified Reference 39, but assume there is more. Data Spaces are imagined as some information snippets distributed that could be accessed via semantic interoperability. The author is mainly confusing data spaces with the open data movement supported by the advent of Linked data. A lot of assumptions are just slightly wrong, but in the sum of them leads to a very confused picture. Many assertions are plainly wrong, like "data linked in a semantic interoperable environment is difficult to fully isolate from other data in that environment for each processing operation. One can guess what that means, but linking the policy data to the payload data is one of the goals of semantically loading data in data lakes. And telco "metadata" and semantic web "metadata" have a very different meaning. Assuming the first one was meant, it is mixed into the text in a way that is hardly understandable to anyone. The separation of identification properties from other data is a typical challenge for anonymization techniques(differential privacy and the like). Those are not affected by "linking". The path becomes a dead end. If the second is meant, the isolation doesn't make any sense or is even counterproductive as was proven by the cited S.Kirrane.

The assertion that GDPR only focused on syntactic interoperability is contradicted by all the discussions during the making of GDPR, the developments around Do-Not-Track. It is even contradicted in the text, Art21 V GDPR. By inventing the principle of separation, the author makes things worse and puts himself into a position where there is no way out anymore. The misunderstanding of data properties qualifying data as personal data with linking, data spaces and the semantic web creates a semantic swamp that is impossible to escape from. The lack of clean goals, scope and definitions make the article very hard to read. The lack of use of the terms commonly used in the field makes it even harder.

From the support statement, one can only guess that the "data space" encompasses solid pods that have access control, policy and other means to share information. Again, a clear scope is a precondition for a comprehensible article in this field.

While recognizing that there was a lot of effort invested in the paper, the insights from the paper are hidden in cloudy and winding wordings. The author discovers the complex field of privacy and the web of data and let us participate in this discovering journey. There are some good questions raised, but the lack of clarity, scope and the many wrong assumptions unfortunately do not allow for those good questions to be developed, let alone to start evaluating the already existing answers to most of them.

The paper and its all encompassing scope MUST lead to a failure, at least in the field it is addressing. The author is strongly encouraged to select very specific aspects first and then try to get the overall picture. Especially in privacy related matters, the overall picture is too complex to start with.

---

### Author Response · Authors · 2024-08-11
**Rationale behind additional modifications**

First of all, the author would like to thank the reviewers for their expertise. Given the specific topic and the limited knowledge on this in legal scholarship, it was very valuable. Posting the PDF in graphics format was not intentional, hopefully this is now OK. Apart from only minor changes in the last two sections, the text was thoroughly rewritten and reworded. In addition, the following changes were made:

-	Although developments on the semantic web and open linked data can be related to developments within European Data Spaces, they are now mainly taken out of the article and the text should therefore be clearer. Now, the distinction should not have to be made over and over again. Any implications and reference to this have also been taken out.
-	Although no unified definition of a Data Space exists, the author has tried to clarify and provide context as much as possible. It was by no means intended to make too much connection with Solid Pods, this has also been remedied now normally.

-	The initial intention of the article is to make known a basis for lawyers and legal scholars, to give an introduction to basic concepts in terms of the broader evolution of data sharing that is set out. Hence, the choice was made to try to use different jargon at times. This has now been adjusted again to try to minimize confusion. The author hopes that a debate at the workshop can further enhance this process.
-	Several nuances were made as well as ambiguities removed regarding the scope of the article. This should make it clear that the main objective is to initiate and fuel an interdisciplinary debate. This in the light of the proposed Designing-by-Debate approach.
-	The principle of separation was further explained and reference was made to the specific author and passages that presuppose this.
-	All references and cross-references were checked again. Several references were removed and new references were added.

---

> ### Comment · Reviewer_cnTx · 2024-08-12
> **Thanks for the clarifications**
>
> Given the explanations above, the goal of the paper becomes much clearer. A review is always a function of the goal expressed, implied or understood. My review was in light of functional novelty of GDPR compliance in dataspaces. Given this is an impulse for an interdisciplinary discussion, I appreciate the new clarity on the goal. I think that the paper is fit for purpose for that goal. I therefore explicitly support the decision of the organizers to accept the paper.

---

### Comment · Program_Chairs · 2024-08-16
**Acceptance notification**

Dear Michiel,

Thank you again for submitting your paper, {{submission_title}}, to NXDG 2024. We are delighted to inform you that your submission has now been accepted for presentation and publication. Congratulations!

At least one author of the paper needs to register for the SEMANTiCS conference to present the paper. We will let you know the presentation format in a follow-up email.

Best,
NXDG 2024 Program Chairs

---

### Decision · Program_Chairs · 2024-08-02

Conditionally Accepted